



# Empirical classification of dry-wet snow status in Antarctica using multi-frequency passive microwave observations

Marion Leduc-Leballeur[1], Ghislain Picard[2], Pierre Zeiger[2], and Giovanni Macelloni[1]

[1]Institute of Applied Physics "Nello Carrara", National Research Council, 50019 Sesto Fiorentino, Italy
[2]UGA, CNRS, Institut des Géosciences de l'Environnement (IGE), UMR 5001, 38041 Grenoble, France

**Correspondence:** Marion Leduc-Leballeur (m.leduc@ifac.cnr.it)

**Abstract.**

Passive microwave satellite observations are commonly used to detect liquid water in the snowpack on the ice sheet. Typically, algorithms yield a binary dry-wet indicator limiting the information. Theoretical analyses have been demonstrated that these dry-wet indicators correspond to different levels in the snowpack depending on the frequency: from surface to ∼0.2 m at 37 GHz, from surface to ∼1 m at 19 GHz and from surface to depths exceeding 1 m at 1.4 GHz. In this study, our objective is to enhance understanding of melting and refreezing processes in Antarctica. For this, we proposed an empirical method that combines several binary dry-wet indicators computed at three frequencies (1.4, 19, and 37 GHz) and for two acquisition times (afternoon/night). We also introduced another indicator to estimate if most of the pixel (> 80 %) is subject to melt. By combining these six binary indicators, we obtained 64 possible daily "dry-wet signatures", which were interpreted to infer whether the snowpack was dry, actively melting, or only wet below the surface, if night refreezing was occurring, and if a large proportion of the pixel was impacted. 98% of the examined pixels show a coherent and physically meaningful daily dry-wet signature across Antarctica during the 2012-2023 considered period. To synthesise the 64 dry-wet signatures, we grouped the signatures conveying similar information into 10 qualitative classes of "snowpack status". This new classification reveals a clear relationship between the various snowpack status and average surface temperature from ERA5 reanalysis, demonstrating the reliability of the empirical definition of the 10 classes. Furthermore, the classification captures the expected seasonal melt evolution: night refreezing is frequent at the beginning of the melt season, while sustained melting is observed in the middle of the summer, and remnant liquid water at depth features the end of the melt season. In the Antarctic Peninsula, over 11 years, we found an increasing trend in melting, significantly related to an increase in remnant liquid water at depth and a decrease in nighttime refreezing. This new classification offers deeper insights in melt processes for investigating extreme events and climate variations compared to previous binary indicators.

## 1 Introduction

The detection of surface melting on the ice sheets by space-borne microwave radiometry has a long history (Zwally and Gloersen, 1977). Numerous melt datasets have been built from these observations and have been used in climate studies of the polar regions, for example to reveal interannual trends or the relationship with other climatic indicators (e.g. Liu et al., 2006;



Picard et al., 2007; Tedesco, 2007; Kuipers Munneke et al., 2012; Nicolas et al., 2017; Wille et al., 2019; Datta et al., 2019; Banwell et al., 2021; Johnson et al., 2022; Kittel et al., 2022; Saunderson et al., 2022; Banwell et al., 2023; Gorodetskaya et al., 2023; Dethinne et al., 2023; de Roda Husman et al., 2024). The detection of surface melting is relatively straightforward because the snowpack thermal emission in the microwave domain radically changes when meltwater appears in the snow matrix (Chang and Gloersen, 1975). It is also noteworthy that microwave frequencies are sensitive to the presence of liquid water,

independently of the fact that snow is actually melting or not. Thus, the detection methods usually provide a binary indicator of the presence or absence of liquid water, i.e., the dry-wet snow status, a variable defined by the World Meteorological Organization (cf. https://space.oscar.wmo.int/variables/view/snow_status_wet_dry, last access: 28 January 2025). Most often, observations at 19 GHz in horizontal polarisation are used because the amplitude of the brightness temperature variation is maximized between wet and dry states (Zwally and Fiegles, 1994; Torinesi et al., 2003). A few combinations of frequencies,

polarisations, day/night overpasses time were also tested (Abdalati and Steffen, 1997; Zheng et al., 2018).

The inception of L-band radiometry in space through the Soil Moisture and Ocean Salinity (SMOS) satellite in 2009 (Kerr et al., 2001) and the Soil Moisture Active Passive (SMAP) satellite in 2014 (Entekhabi et al., 2010) has opened up a new opportunity to detect melt. The algorithms previously developed for higher frequencies have proven effective at L-band as well. However, the information content of the resulting dry-wet status differs significantly (Leduc-Leballeur et al., 2020). L-

band is indeed characterised by a very low absorption of the microwaves in the dry snow (Mätzler, 2006; Passalacqua et al., 2018), allowing microwaves to emerge from depths up to hundreds of meters when water is completely absent. In principle, it enables the detection of buried meltwater even when the surface is refrozen. This unique characteristic has recently been also exploited in Greenland to detect perennial firn aquifers with SMAP (Miller et al., 2020) and to estimate the among of liquid water with SMOS (Houtz et al., 2021). This new perspective highlights the different information retrieved depending on the

frequency.

Recent studies specifically investigated the sensitivity to the presence of meltwater as a function of frequency, especially with respect to the depth of detection. In Colliander et al. (2022), liquid water content observations at different depths up to 4 m at the DYE-2 experimental site in Greenland were correlated to the microwave signal at multiple frequencies. The data show how the melt season unfold, from initial surface melting to the percolation and refreezing of meltwater at depth, and how

the microwave signals at the different frequencies follow these different stages. In Picard et al. (2022), a modeling approach is taken to compute the theoretical maximum depth of detection for a given frequency in a typical Antarctic snowpack. Whilst both studies yield different values for these depths, they both showed that frequencies lower than 19 GHz are sensitive to water at gradually greater depths. Conversely, at 37 GHz, the sensitivity is limited to a shallower zone under the surface, definitely invalidating the term "surface melting" loosely used in the past to refer to outputs of the 19 GHz-based algorithms

(e.g. Zwally and Fiegles, 1994; Torinesi et al., 2003; Tedesco et al., 2007; Picard and Fily, 2006; de Roda Husman et al., 2023). Theses two studies illustrate the potential to provide more insight on melt processes from the large frequency range available by contemporary radiometric missions, and expected from future missions (e.g., Copernicus Imaging Microwave Radiometer (CIMR, Donlon, 2023); Advanced Microwave Scanning Radiometer 3 (AMSR3, Kachi et al., 2023)). Based on these recent findings, Colliander et al. (2023) used passive microwave observations between 1.4 GHz and 36.5 GHz available from SMAP



and Advanced Microwave Scanning Radiometer 2 (AMSR2) to monitor surface and subsurface meltwater vertical distribution over Greenland. Nevertheless, these recent studies highlight how difficult and uncertain are current estimations of liquid water content and wet layer depth.

In this study, we adopt an intermediate approach between the binary dry-wet detections, which have proven their usefulness in numerous polar climate studies, and the recent attempts aiming to estimate the quantity or the depth of melt. We propose an empirical classification of the dry-wet snow status, exploiting the combination of multi-frequency observations from AMRS2 and SMOS (1.4 – 37 GHz) to provide enhanced information on melt processes. Our algorithm provides for each pixel and each day the snowpack status in 10 classes from "dry" to "all day full melting" via "wet at depth without melting" and other intermediate stages. The algorithm also gives some qualitative information on the uncertainties and when the combined wet-dry indicators provide inconsistent information. We ran this algorithm on the Antarctic ice sheet from 2012 to 2023 at 12.5 km resolution. Noting that a strict validation of our dataset is impossible due to the lack of adequate in situ measurements, and following the approach of other studies (e.g. Torinesi et al., 2003; Colliander et al., 2023), we performed some comparisons with air surface temperature and assessment of the seasonal variations to check their physical consistency. At last, we investigated the climatic information content of this new dataset by exploring trends and seasonal and interannual variations.

## 2 Data sets

### 2.1 Brightness temperature observations

#### 2.1.1 AMSR2 observations at 19 and 37 GHz

Brightness temperature at 19 and 37 GHz were obtained from the Advanced Microwave Scanning Radiometer 2 (AMSR2) on-board the Japan Space Agency (JAXA)'s Global Change Observation Mission 1 – Water "SHIZUKU" (GCOM-W1) satellite. The AMSR-E/AMSR2 Unified Level 3 daily product version 2 processed by the National Snow and Ice Data Center (NSIDC; Maslanik and Stroeve, 2004, updated 2018; https://nsidc.org/data/au_si12/versions/1, last access: 25 March 2024) is used here. This product provides the daily mean brightness temperatures acquired during all the ascending and descending passes respectively, projected onto the southern polar stereographic projection (ESPG: 3976) with a resolution of 12.5 km at vertical and horizontal polarisations for an incidence angle of $55^o$. In Antarctica, the ascending passes occur from 13:00 to 17:00 (afternoon) and the descending passes from 21:00 to 01:00 (night), local time, which enables capturing the diurnal variability (Zheng et al., 2018). However, there is a technical difficulty due to how the daily mean is calculated by the data provider. The passes are grouped by "day" according to UTC time, that is from 00:00 to 23:59 UTC, irrespective of the local time. This has negative consequences for the interpretation of our dataset, with two levels of severity. In the most favourable cases, all ascending passes in a UTC day are acquired from successive orbits, within a few hours, and are all grouped either before or after the group of descending passes. In this favourable situation, the local afternoon observations are before the following night or before the preceding night. In the worst situations, the average descending passes encompass acquisitions from two distinct nights. This situation occurs within the Atlantic sector around longitudes $\sim0^o$ (UTC+00 zone). Symmetrically, the ascending





pass average may contain acquisitions from two different days in the pacific sector around $180^o$ longitudes (UTC+12 zone). These worse cases represent 10 % of the pixels according to our evaluation using the acquisition time recorded in the Level 3 product provided by the JAXA's Globe Portal System (G-Portal; https://gportal.jaxa.jp, last access: 25 March 2024). This problem can not be solved without reprocessing all low level data. Meanwhile, it implies a cautious and flexible interpretation of "day" and "night" in the following. In practice the issue is certainly minor for climate investigations (e.g., climatological occurrence of day versus night melt), but becomes critical when investigating a precise sequence of meteorological events, such as the impact of an atmospheric land-fall, which evolves at hourly time scales (e.g. Wille et al., 2021).

### 2.1.2 SMOS observations at 1.4 GHz

Brightness temperature at 1.4 GHz is obtained from the European Space Agency (ESA)'s Soil Moisture and Ocean Salinity (SMOS) satellite collaboratively developed with the Centre National d'Études Spatiales (CNES, France) and the Centro para el Desarrollo Tecnológico Industrial (CDTI, Spain). We used the SMOS enhanced resolution product of brightness temperature built by Zeiger et al. (2024) (last access: 05 December 2024, Zeiger and Picard, 2024) on the southern polar stereographic projection (ESPG: 3976) with a 12.5 km resolution. This product provides the brightness temperature at vertical and horizontal polarisations and $40^o$ incidence angle with an effective spatial resolution of about 30 km. This number is twice better than the native SMOS observations ($\sim$40–70 km). It is closer to the spatial resolution of AMSR2 products (about 20 km at 19 GHz and 10 km at 36.5 GHz), which makes it suitable for input in a multi-frequency algorithm. The SMOS daily timeseries are obtained by averaging the mean morning and afternoon brightness temperatures corresponding approximately to the ascending and descending SMOS passes, respectively.

### 2.2 Skin temperature

Skin temperature was used to assess the coherency of the classification. It is taken from the European Centre for Medium-Range Weather Forecasts (ECMWF) Reanalysis v5 (ERA5) downloaded from the Copernicus Climate Change Service (C3S) (Hersbach et al., 2018, last access: 25 March 2024). ERA5 provides data in hourly temporal resolution and covers Antarctica in a regular latitude-longitude grid of $0.25^o \times 0.25^o$ (Hersbach et al., 2020). This dataset was projected on the southern polar stereographic grid and interpolated at a resolution of 12.5 km using nearest neighbours.

### 3 Method

The classification algorithm developed in this study proceeds in three main steps: 1) compute binary dry-wet snow status, called dry-wet snow indicator, at three frequencies (1.4, 19 and 37 GHz) quasi independently; 2) combine them to obtain a more elaborated description of the dry-wet snow status, called dry-wet signature, each one elucidated in physical terms; 3) group together in the same class the dry-wet signatures sharing physical interpretations to form 10 distinct classes, called snowpack status, with corresponding physical explanations, for every given UTC day and in every pixel independently. A land/sea mask is applied to eliminate sea and mixed pixels. The algorithm steps are depicted in Fig. 1 and described below.





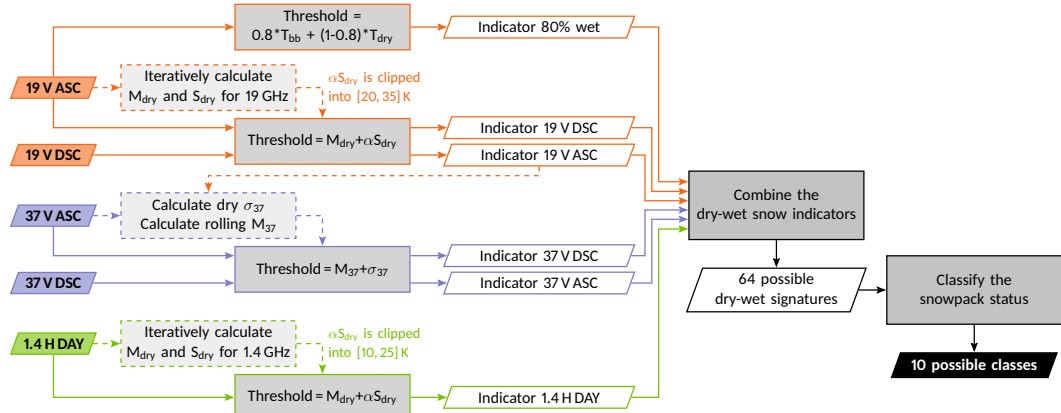

**Figure 1.** Process diagram of the dry-wet snow classification. Dashed arrows are pre-processing steps to compute the brigthness temperature thresholds.

## 3.1 Single frequency dry-wet snow indicators

The 19 GHz-based dry-wet snow indicator follows the Torinesi et al. (2003) algorithm inspired by Zwally and Fiegles (1994).

The algorithm determines an optimal brightness temperature threshold for every melt year (defined as 1 April year N to 31 March year N+1) in every grid cell and considers that any acquisition of brightness temperature higher than this threshold indicates wet snow. The main challenge is to find an adequate threshold. Torinesi et al. (2003) proposed a simple adaptive method in which the brightness temperature threshold is: $T_{BT} = M_{dry} + \alpha S_{dry}$ where $\alpha = 3$, and $M_{dry}$ and $S_{dry}$ are the mean and standard deviation of the brightness temperature timeseries when snow is detected as dry (called "dry days" hereinafter,

and symmetrically for the wet status). To solve the circular problem of computing $M_{dry}$ and $S_{dry}$ for dry days in order to detect wet days, the initial step calculates the all-day mean brightness temperature $M_{dry}^{(0)}$ for every melt year, and for each grid cell independently and set $\alpha S_{dry}^{(0)}$ to a fixed value (10 K). Using this rough threshold, the algorithm computes a first timeseries of snow status, which is then refined by multiple iterations of the same process. The convergence is reached after three iterations (Torinesi et al., 2003). After observing that the brightness temperatures acquired from ascending or descending passes are

always very close to each other during the dry season (<1.3 K on average over 2012-2023), the threshold is determined only with the ascending passes (more subject to melt), and is then applied to both the ascending and descending passes timeseries.

We implemented the Torinesi et al. (2003) method with two improvements. First, we used the vertical polarisation as suggested by Picard et al. (2022) and used in Colliander et al. (2023). The vertical polarisation offers a more stable signal during dry conditions with respect to the horizontal polarisation, which is more sensitive to surface density and stratification of the

snowpack, and thus more subject to snow metamorphism variations and local peculiarities. Second, noting that a too low threshold was generating false alarms (especially obvious in the winter) and a too high one reduces the sensitivity of the algorithm, we added upper and lower bounds to the threshold by limiting $\alpha S_{dry}$ inside the 20 – 35 K range.



The choice of $\alpha$=3 is typical for outlier detection (e.g. von Storch and Zwiers, 2001) and has been confirmed to perform well for melt detection after throughout investigation of the brightness temperature timeseries (Torinesi et al., 2003). Here, in
addition, two other values (2.5 and 3.5) have been considered to assess the impact of this parameter and quantify an uncertainty range.

The detection of melt using 1.4 GHz observations is based on the 19 GHz algorithm with some adaptation proposed by Leduc-Leballeur et al. (2020), to take into account the lower sensitivity to liquid water at L-band (Picard et al., 2022). The threshold $T_{BT}$ is computed from the daily brightness temperature at 1.4 GHz in horizontal polarisation with a first-guess equal
to 15 K. Furthermore, $\alpha S_{dry}$ is limited inside the $10 - 25$ K range. Moreover, to reduce false detections, pixels with an annual standard deviation of brightness temperature in vertical polarisation lower than 2.8 K are filtered out (Leduc-Leballeur et al., 2020). This filter removes numerous false alarms in the interior of the ice sheet where melt is obviously absent.

Picard et al. (2022) highlighted the potential to distinguish different stages of surface melt from 37 GHz. Brightness temperature simulations showed that a dry 10 cm thick snow layer with coarse grains (refrozen crust) over a wet snowpack is detected as
dry at 37 GHz and as wet at 19 GHz. This characteristic makes 37 GHz suitable for the detection of near surface melt and night refreezing that is usually limited to the topmost centimeters of the snowpack. From this analysis, an indicator was developed to provide information on the dry-wet surface status. Nonetheless, the method previously used for melt detection at 19 GHz and 1.4 GHz is inadequate at 37 GHz due to its strong sensitivity to variations in snow grain size.A new threshold definition was adopted: $T_{37} = M_{37} + \sigma_{37}$ where $M_{37}$ is the 5-day moving mean timseries of the brightness temperatures when the 19 GHz
indicator is dry and $\sigma_{37}$ is its standard deviation between 1 April year N to 31 March year N+1. $T_{37}$ is computed from the brightness temperature in vertical polarisation acquired at ascending passes and subsequently applied to both ascending and descending passes. This threshold still exhibits a strong sensitivity to brightness temperature variations, leading to occasional unexpected melt detection during winter (e.g. 296 pixels in July-August on average, i.e. 0.13 % of the total wet days detected with 37 GHz). These false alarms underscore that melt detection at this frequency remains difficult and subject to uncertainties.
Finally, we established an indicator to qualify if most of the pixel is wet. For that, we designed a threshold based on the 19 GHz brightness temperature acquired in ascending passes to match to approximately 80 % of the pixel with wet snow, as follows: $T_{80\%} = 0.8 * T_{bb} + (1 - 0.8) * T_{dry}$. $T_{bb} = 273\,K$ is the theoretical maximum brigthness temperature during melting (black body) at V polarisaion near the Brewster angle (Picard et al., 2022). $T_{dry} = 205\,K$ is the mean brigthness temperature of the dry snow computed in July-August where more than 10 days are observed by year on average over 2012-2023. In these
conditions, $T_{80\%} = 260\,K$ and the indicator is set to 1 when the 19 GHz brightness temperature acquired in ascending passes exceeds it, otherwise it is set to 0. In the following, we call "partial melting" pixels with less than 80 % of melt and "full melting" otherwise.

## 3.2   Dry-wet signatures

To compute the dry-wet signature, we combined the dry-wet snow indicators at three frequencies (1.4, 19 and 37 GHz), with the
separation of the ascending/descending passes for AMSR2 (the two highest frequencies), and the sixth indicator of "partial/full melting". This resulted in $2^6 = 64$ possible daily dry-wet signatures that we interpreted in physical terms and for which a digit





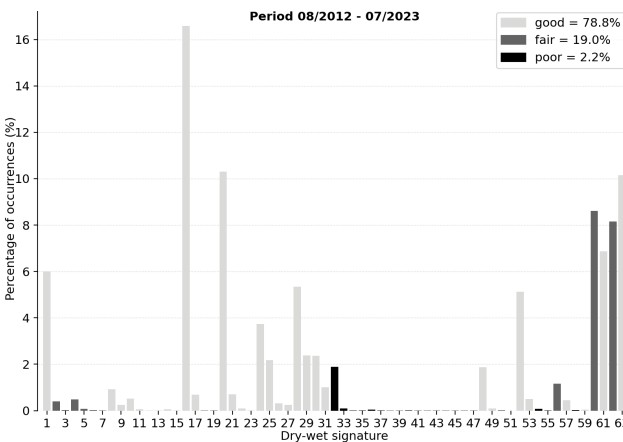

**Figure 2.** Percentage of each dry-wet signature, excluding the dry days (signature 0), between August 2012 and July 2023 in Antarctica. Their respective quality flags are indicated as grey scale.

is attributed for identification (cf. Appendix A). Based on the theoretical analysis by Picard et al. (2022), we interpreted the 37, 19 and 1.4 GHz observations as the dry-wet snow status in the topmost 0.2 m, 1 m, and > 1 m of the snowpack respectively. Note that in case of active melting in upper surface, the three frequencies become sensitive only to few topmost wet centimeters

of the snownpack and no information is available below (Picard et al., 2022). Thus, the dry-wet signatures are qualitatively described by using "surface" or "deep" without attempt to quantify the liquid water profile (cf. Appendix A). The descending passes provide information on the occurrence of night refreezing at the surface (37 GHz) or at depth (19 GHz). The "partial/full melting" indicator reinforces the reliability of the melt detection by assessing when more than 80 % of the pixel is affected by melt.

In addition, the consistency level of each signature was qualified with a quality flag: 2 (good) is assigned when all indicators depict a physically meaningful snowpack status, 1 (fair) when one or two indicators are inconsistent, and 0 (poor) when one or more indicators are severely inconsistent (cf. Appendix A). For instance, the signature 47 corresponds to all the dry-wet indicators equal to 1 except the ascending 19 GHz indicator equal to 0. It means that the majority but not all the indicators are in agreement and the signature is therefore flagged as fair. In contrast, the dry-wet signature with a brightness temperature

exceeding 260 K at 19 GHz for ascending passes but all the other indicators equal to 0 (signature 32) is considered inconsistent and flagged as poor. The situations leading to fair and poor qualities may be due to the erroneous detection in one indicator, the difference of sensor spatial resolutions, or uncommon timing of the wet snow occurrence.

Figure 2 illustrates the percentage of each dry-wet signature among days detected as wet (hence excluding the "dry" signature, signature 0) from August 2012 to July 2023. Over this period, the 9 most frequent signatures (occurring more than

5 % individually) contribute to 77.1 % of the time all together. Conversely, the 35 less frequent signatures (occurring less than 0.1 % individually) contribute to 0.8 % of the time. Over the whole period, the obtained wet signatures are mostly qualified as physically coherent (78.8 %), or fair (19.0 %) and only 2.2 % are poor.





### 3.3 Empirical snowpack status classification

Each signature was further assessed and grouped into one of 10 physical descriptions of the snowpack state (Table 1). This
classification is defined empirically and, although the descriptions are arguably subjective, it helps reduce the complexity of
the 64 dry-wet signatures into only 10 classes. One color was assigned to each class by selecting similar shades for classes
with a close interpretation. This enables us to offer a quick overview while maintaining the distinctions between the detailed
classes through color variations. The 10 classes are defined as follows.

The class "all day dry" (class 0) includes the signature where all indicators are zero, and also signatures where only the
37 GHz indicator (ascending or descending) is one. Although the latter may suggest superficial melting, we prioritize the
agreement between the 19 and 1.4 GHz indicators over the 37 GHz indicator over the limited reliability of the 37 GHz detection
algorithm during winter. The class "wet at depth without melting" (class 1) is led by signatures with the 1.4 GHz indicator at
one and the 19 GHz indicator at zero, meaning that surface is dry, but liquid water is present below in the snowpack. Note that
the depth of detection for each frequency is uncertain and is sensitive to the particular snowpack conditions in each location
and year (Colliander et al., 2022; Picard et al., 2022).

Five classes are assigned to partial melting with differences related to their night refreezing status. The classes "daytime par-
tial melting with night refreezing" (class 2) and "daytime partial melting with night surface refreezing" (class 3) are determined
by the 19 GHz indicator equal to 1 in the ascending pass (i.e. daytime) and, respectively, the 19 and 37 GHz indicator equal to
0 in the descending pass (i.e. nighttime). The class "wet snow with uncertain surface" (class 4) includes signatures indicating
215   the presence of liquid water based on the 19 GHz indicators, but with discrepancies in the 37 GHz indicators, bringing uncer-
tainty regarding active melting at the surface. The class "all day partial melting" (class 5) refers to signatures with the 37 GHz
indicator in both passes equal to 1 and at least one 19 GHz indicator equal to 1. The class "nighttime partial melting" (class 6)
includes signatures with the 19 GHz indicator equal to 0 in the ascending pass but one in the descending one, regardless of the
37 and 1.4 GHz indicators.

Three classes are defined as full melting referring to the indicator based on the brightness temperature at 19 GHz acquired
at ascending passes above 260 K, i.e. for which over 80 % of the surface pixel is affected by melt. The differences are linked
to their refreezing status. The classes "daytime full melting with night refreezing" (class 7) and "daytime full melting with
night surface refreezing" (class 8) are determined by the 19 GHz indicator at one in the ascending pass (i.e. daytime) and,
respectively, the 19 and 37 GHz indicator at zero in the descending pass (i.e. nighttime). The class "all day full melting"
includes signatures with the 37 GHz indicator in both passes at one and at least one 19 GHz indicator at one.

Finally, the class "invalid" (class -1) includes the signatures with a lack of physical coherency, for which the brightness
temperature at 19 GHz is above 260 K but the 19 GHz indicator acquired at ascending passes is zero. This may be due to
a threshold detection exceeding 260 K where brightness temperature remains relatively high (about $> 230$ K) during some
winter. de Roda Husman et al. (2023) already identified that the threshold method tends to underestimate melt over persistent
melting regions.





Two classes have a more ambiguous interpretation. First, the class "wet with uncertain surface status" (class 4) has uncertainty mainly coming from the 37 GHz melt indicator, which stems from the difficulty of using 37 GHz to detect melt. Second, the class "nighttime partial melting" (class 6) has uncertainty due to the time reference issue of the AMSR2 product raised in Sect. 2.1.1, but it may also correspond to real situations, such as when warm air is advected by atmospheric synoptic events

with land-falling during the night rather than the day before. We advise future users of the dataset to consider this possibilities in their statistical analysis and explore other meteorological indicators (e.g. synoptic charts) to consolidate the interpretation of this class.

**Table 1.** The 10 daily snowpack classes and their matching signatures (cf. Fig. A1) and colors. Italic face indicates the rare signatures, defined as with less than 250 occurrences ($< 0.01$ %) from August 2012 to July 2023 over Antarctica.

| Class | Daily snowpack status | Associated dry-wet signatures |
|---|---|---|
| -1 | Invalid | 32, 33, *34*, *35*, 36, *37* |
| 0 | All day dry | 0, 2, 4, 6 |
| 1 | Wet at depth without melting | 1, 3, 5, *7* |
| 2 | Daytime partial melting with night refreezing | 16, 17, *18*, *19*, 20, 21, 22, 23 |
| 3 | Daytime partial melting with night surface refreezing | 28, 29, *44*, *45* |
| 4 | Wet with uncertain surface status | 24, 25, 26, 27, 40, *41*, *42*, *43*, 56, 57, 58, 59 |
| 5 | All day partial melting | 30, 31, *38*, *39*, *46*, *47* |
| 6 | Nighttime partial melting | 8, 9, 10, 11, *12*, *13*, 14, 15 |
| 7 | Daytime full melting with night refreezing | 48, 49, *50*, *51*, 52, 53, 54, 55 |
| 8 | Daytime full melting with night surface refreezing | 60, 61 |
| 9 | All day full melting | 62, 63 |

## 3.4 Sensitivity of the classification to the detection threshold

To assess the stability of the classification algorithm, the impact of variations in the $\alpha$ parameter of the 19 GHz detection

algorithm is estimated. Two confusion matrices comparing the standard detection ($\alpha = 3$) with the cases $\alpha = 2.5$ and $\alpha = 3.5$ respectively are presented in Fig. 3. In the case of a lower threshold detection ($\alpha = 2.5$), an agreement higher than 92 % is observed for 7 classes, suggesting only little sensitivity to decreasing $\alpha$. In contrast, the highest sensitivity is observed for the class "nighttime partial melting" with 22 % conversion into the class "wet with uncertain surface status". Similarly, about 11 % of the class "daytime full melting with night refreezing" (class 7) is converted into the class having night refreezing limited

to the surface (class 8). Finally, about 22 % of dry-wet signatures identified as lacking in physical explanation (class -1) are converted into full melting with night refreezing (class 7) by using a lower threshold detection due to the class -1 definition.

In the case of a higher threshold detection ($\alpha$=3.5), 7 classes are also fairly insensitive with agreements higher than 93 %. The class "nighttime partial melting" is again the most sensitive with 36 % conversion into no active melting presence (classes



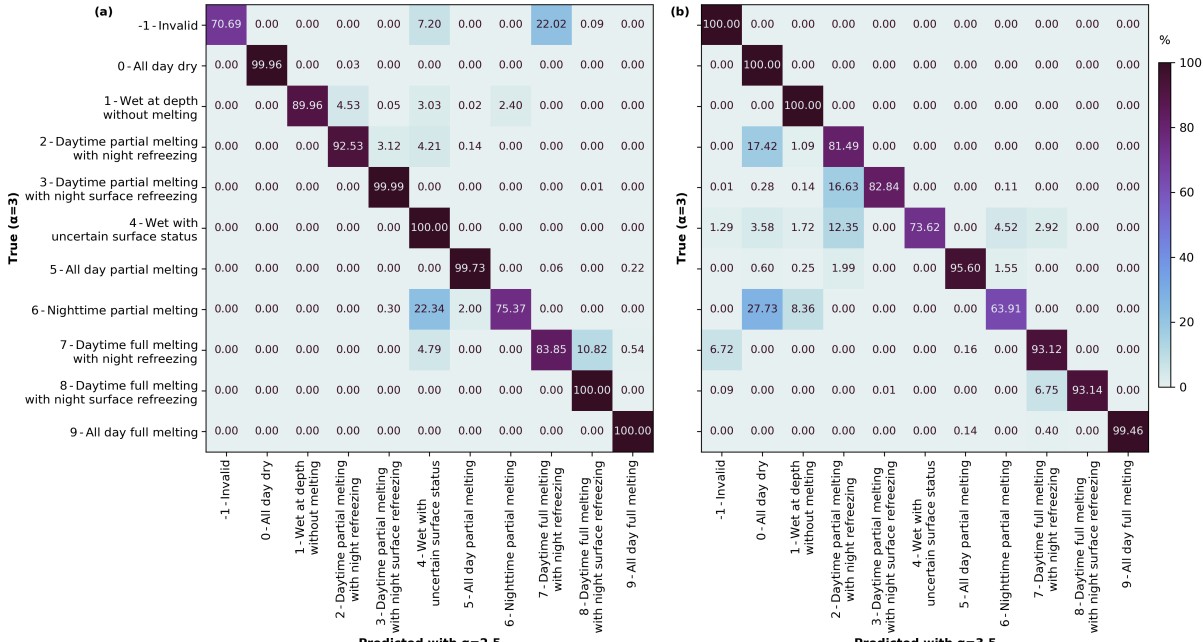

**Figure 3.** Confusion matrix computed between the classification with $\alpha$=3 (true condition) and predicted with (a) $\alpha$=2.5 and (b) $\alpha$=3.5. Results are normalized over the true condition.

0 and 1). 17 % of the class "daytime partial melting with night refreezing" (class 2) is also converted into "all day dry" (class 0).

250   Due to this higher threshold, night refreezing is more frequently detected (from classes 3 and 4 into class 2 for 17 % and 12 % respectively, and from class 8 into class 7 for 7 %). Lastly, the class "wet with uncertain surface status" (class 4) is converted mainly into nighttime partial melting (5 %), partial melting with night refreezing (12 %) and dry (4 %).

In summary, variations in $\alpha$ at 19 GHz has an overall small impact on the classification, and the main impact is on the night refreezing status. The most affected classes are "nighttime melting" (class 6) and "wet with uncertain surface status" (class 4).

255   Future users of the dataset interested by these most affected classes should consider using the three $\alpha = 2.5, 3, 3.5$ to assess the robustness of their investigation.

## 4   Results and discussion

### 4.1   Comparison to ERA5 temperature

We compare the classification dataset to the ERA5 skin temperature. While a direct validation of microwave melt detection with

260   in situ observations is inherently impossible (van den Broeke et al., 2023), a few comparisons to air temperature measurements from the Automatic Weather Station (AWS) have usually highlighted good consistency (e.g. Torinesi et al., 2003; Colliander





et al., 2023; de Roda Husman et al., 2023) despite a large difference in spatial representation between AWS and satellite pixel. Here, we use the ERA reanalysis to benefit from its coverage over the whole continent.

Figure 4 depicts the distribution of daily minimum and maximum temperature occurrence for each class. This distribution is obtained for all pixels over Antarctica and all days from 2012 to 2023. The dry status (class 0) presents a mean maximum temperature of -32.8±15.0$^o$C and minimum of -38.3±14.3$^o$C. On average, the presence of wet snow at depth without melting (class 1) is associated with the lowest minimum and maximum temperatures (-12$^o$C and -5$^o$C respectively), which supports the absence of liquid water detection at the surface. The mean maximum temperature in classes 2, 3, 4 and 5 increases from -3$^o$C to -1$^o$C while the mean minimum temperature increases from -11$^o$C to -6$^o$C, supporting the gradual disappearance of nighttime refreezing when temperature are tending to the freezing point. The nighttime partial melting status (class 6) has a mean minimum temperature (-8$^o$C) higher than the classes with night refreezing (classes 2, 3, 4, 7, 8) and also lower diurnal temperature variations, which is expected. Similarly to classes 2-5, the three "full melting" classes (7-9) present a gradual disappearance of nighttime refreezing that is related to a 3$^o$C increase in mean minimum temperature and a 1$^o$C increase in mean maximum temperature. Moreover, when full melting is detected, the mean maximum temperature increases by 1.0-1.6$^o$C and its standard deviation is reduced by 0.4-0.7$^o$C with respect when the partial melting occurs. Higher temperatures with less variability explain the expansion of melting to more than 80 % of the pixel.

Overall, ERA5 skin temperature supports well the physical meaning of snowpack status and highlight the consistency of the classification. Note that because of the large dispersion in each class and the overlap between different histograms, skin temperature data could not be used directly to derive information on the melting status. The classification presented here is therefore essential to monitor gradual surface refreezing, remanent liquid water, and to distinguish partial and full melting status.

## 4.2 Seasonal variability

Figure 5 presents the timeseries of brightness temperature (color plain curves) at the three frequencies (1.4, 19 and 37 GHz) for three examples. The single frequency dry-wet snow indicators are reported as colored dots for each frequency and the associated daily class is depicted by bar graphs.

The first example is from the Antarctic Peninsula for the 2015/16 melt season (Fig. 5a). From October to mid-December, a few short occurrences (3-6 days) of wet snow are observed, mainly classified in classes 2 and 5, before the main continuous melt period, which lasts more than two months between December and February. Two melt events are also observed at all frequencies in May highlighting the possibility of late autumn melting, which indeed impacts the snowpack at depth according to 1.4 GHz observations. These two events in May 2016 have been already identified in the literature with modeled liquid water up to 2-m in depth Datta et al. (2019).

The second example is located on the Shackleton ice shelf in 2017/18 (Fig. 5b). It illustrates a much shorter melt season (less than two months) with continuous surface night refreezing. 10 days after the season begins, the melt is intense enough to be detected at 1.4 GHz, indicating that liquid water progressively penetrates the snowpack at depth. Remnant meltwater at depth (class 1, gray bars) is observed in February at the end of the wet snow season before a last brief but full melt event.




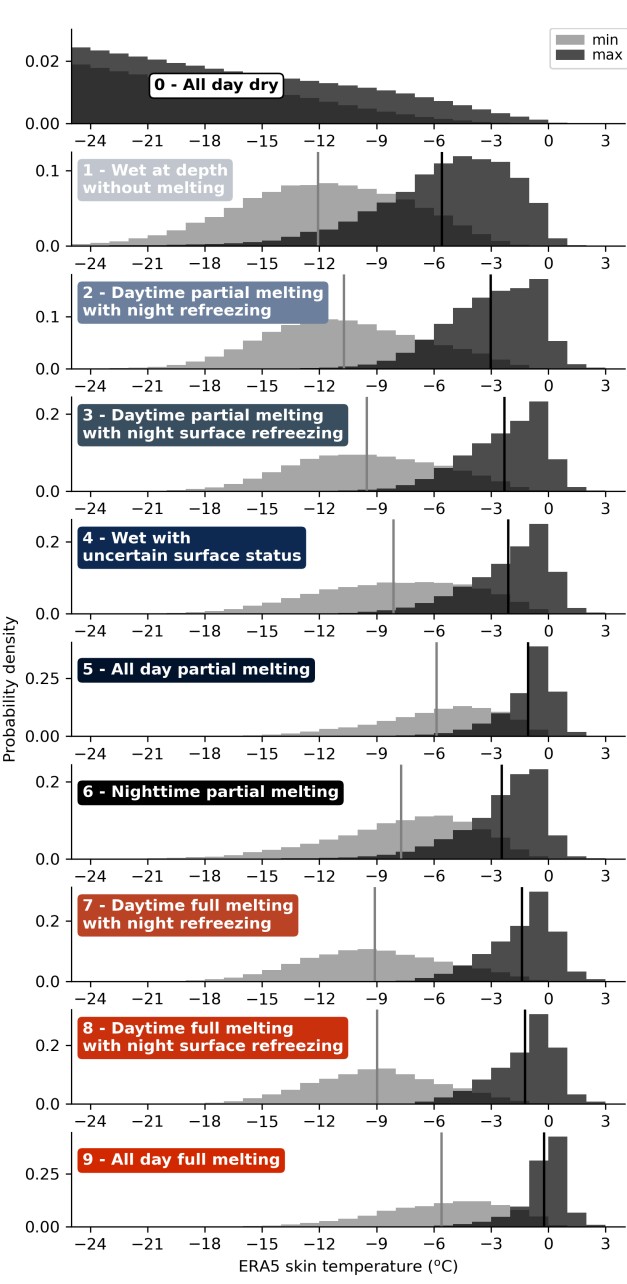

**Figure 4.** Occurrence of the minimum (grey) and maximum (black) daily ERA5 skin temperature for each class from August 2012 to July 2023 over Antarctica. The vertical lines show the mean of the distribution.



The last example is on the Ross ice shelf where only brief and sporadic melt events occur between December 2012 and January 2013 (Fig. 5c). No wet status is detected at 1.4 GHz, suggesting that melt events are not intense enough to affect the snowpack at depth.

In the three examples, the daily maximum surface temperature in ERA5 (Fig. 5 grey lines) is higher than -7 $^{o}$C when melt is detected, and melt is always detected when the surface temperature is higher than the freezing point.

To generalize these examples, Fig. 6 illustrates the surface area of each class in Antarctica throughout the 2012-2023 average melt season. It shows that the melt season typically begins with brief melt events, marked with night refreezing (class 2). From December, the melt quickly spreads over Antarctica (up to $0.6 \cdot 10^6$ km$^2$) and some periods of melting without refreezing (classes 5 and 9) appear and stretch. Nevertheless, throughout the year, the daytime melting with night refreezing is predominant (classes 2, 3, 7, 8 account for 61 %) compared to melt without refreezing (classes 5 and 9 account for 22 %). From the end of January, the extent of melting sharply decreases, and from March, melt occurs only in a few places. The class "wet at depth without melting" (class 1) is rare at the beginning of the melt season, but its occurrence increases from January onwards, representing the highest proportion from mid-February to mid-April. This is in agreement with the seasonal warming of the snowpack, producing meltwater that then percolates and remains present without refreezing (Humphrey et al., 2012). From April, isolated melt events still happen, but wet snow without melting is not detected at depth anymore. This suggests that the seasonal cooling at this stage was sufficient to refreeze the snowpack and the last melt events are too weak to produce and inject a significant amount of liquid water at depth.

In summary, Fig. 6 demonstrates that the classification depicts a consistent unfolding of the typical melt season over the continent.

## 4.3 Spatial and interannual variability

The 2012-2023 averaged number of days per year for each class is depicted in Fig. 7 over Antarctica. Wet snow is confined to the coastal areas and ice shelves, a well-know fact (Zwally and Fiegles, 1994). However, the mean occurrence of each class varies depending on the location. On average, the highest number of days with wet snow occurs in the Antarctic Peninsula, the Dronning Maud Land and the Amery-Shackleton coast. In the Antarctic Peninsula, the most frequent class is "all day full melting" (class 9) with an average of 13 days per year (maximum is 68 days per year on average). It is also where the highest number of days per year with wet snow at depth without melting (class 1, maximum is 46 days per year on average) occurs. Along the Amery-Shackleton coast, the classes of daytime partial melting with night refreezing (classes 2 and 3) and full melting (classes 8 and 9) are the most observed ones, and others classes are rare, with less than 1 day per year on average. The Ross ice shelf area experiences very little melt. On average, the class "wet snow at depth without melt" is rare, and the classes of daytime partial melting with or without night refreezing occasionally happens (0.5 days per year for class 2, and about 0.2 days for classes 3, 4, 5). Few days with full melting are observed (0.7 days per year for class 9).

In the Antarctic Peninsula, the annual occurrence of the full melting classes (7-9) always represents more than 40 % of the total occurrence of the wet classes. In the three other areas, the melt season mainly features daytime partial melting with night refreezing for about 50-60 % (total for the classes 2-3). The Dronning Maud Land and Amery-Shackleton areas are more



**Figure 5.** Brightness temperature at 19 GHz (orange), 37 GHz (purple), 1.4 GHz (green) with wet snow depicted by square markers. Bars and colors show the resulting classification (cf. Table 1 for the color legend). On (a) the Antarctic Peninsula in 2015/16, (b) the Shackleton ice shelf in 2017/18, and (c) the Ross ice shelf in 2012/13. ERA5 daily maximum skin temperature (grey) with temperatures above 0 $^o$C marked in red.



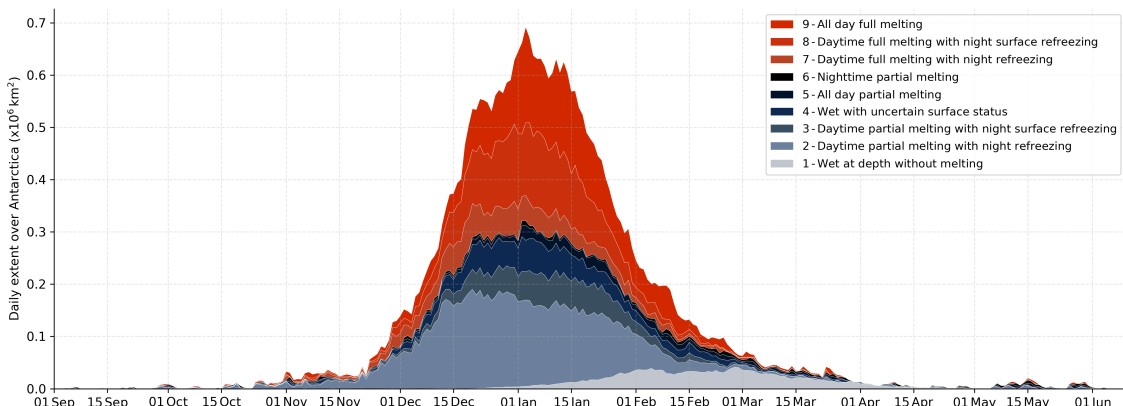

**Figure 6.** Daily extent of each snowpack status over Antarctica from September to June on average over 2012-2023.

subject to night refreezing than the Antarctic Peninsula (classes 2-3 and 7-8), relative to their respective mean annual number of wet days.

Figure 8 highlights the interannual variability for the four main melting areas presented in Fig. 7. The annual occurrence varies by up to 60 % around the mean over all classes. In particular, over the Ross ice shelf where melt is rare (Fig. 8d), the variability is the highest (100 % on average) with an extreme variation in 2015/16 related to a strong El Niño event (Nicolas et al., 2017).

In the Antarctic Peninsula, we find that the annual occurrence of full melting (class 9) and wet snow at depth without melting (class 1) both increase by about 10 % over 2012-2023 and have synchronized interannual variations (significant Pearson correlation $r$=0.85, p-value=0.0010, Fig. 8a, bottom). At the same time, melting with night refreezing (classes 2-3) decreases by about 10 % and the variations are significantly anti-correlated with the presence of wet snow at depth without melting (class 1) ($r$=-0.90, p-value=0.0001). We explain this remarkable results by the fact that when the night refreezing decreases and full melting increases, meltwater is able to percolate and the snowpack remains wet at depth after the period with active melting. A clear significant anti-correlation between class 1 and classes 2-3 is also observed over the Amery-Shakcleton coast ($r$=-0.81, p-value=0.0022, Fig. 8c, bottom), but not with class 9. This could be related to the clear decreasing interannual trend of the full melting, and to the fact that the classes with night refreezing contribute for more than 60 % of the total wet occurrence. The physical consistency of the interannual variations between some classes underscores the reliability of the proposed melt classification.

## 4.4 Discussion and Limitations

The multi-frequency approach to better investigate melt processes has been already proposed by Colliander et al. (2022); Picard et al. (2022); Colliander et al. (2023). These studies highlight the difficulty of quantifying liquid water volume and the depth and thickness of wet snow layers due to the saturation of brightness temperature. As an alternative, we chose here to develop a qualitative classification. The limitation of this approach is that the definition of the terms "partial/full", "surface" or "depth"





**Figure 7.** Annual number of days per year for each class on average over 2012-2023 in Antarctica, the Antarctic Peninsula, Droning Maud Land, along Amery-Shackleton coast and Ross ice shelf. The blue areas are masked out.



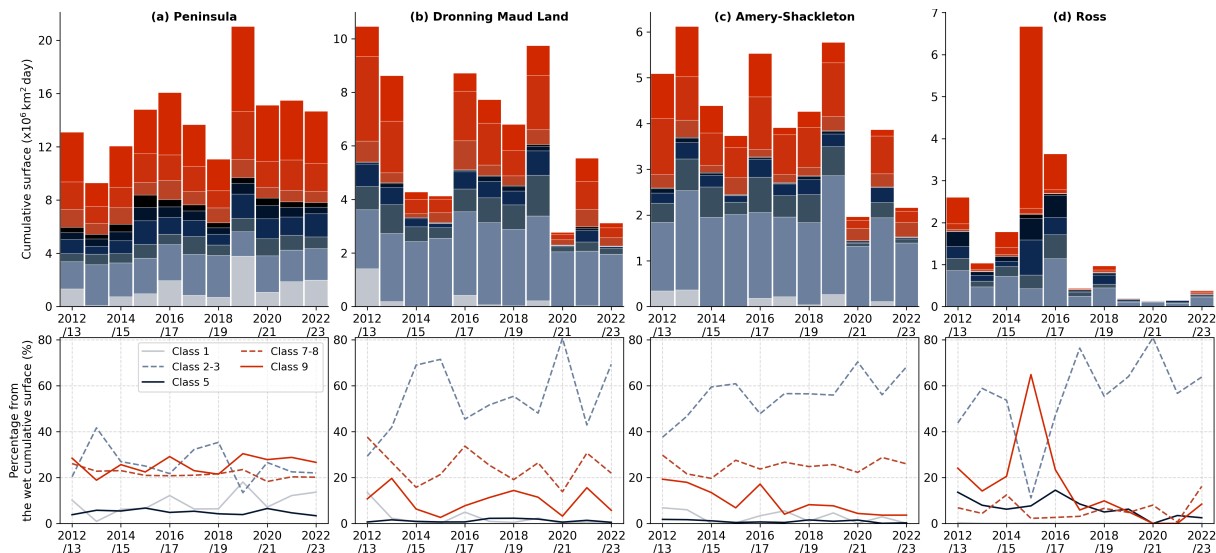

**Figure 8.** (top) Annual cumulative surface for each class from 2012 to 2023 over (a) the Antarctic Peninsula, (b) Droning Maud Land, (c) Amery-Shackleton coast and (d) Ross ice shelf. See Table 1 for the color legend of each class. (bottom) Relative percentage of the class occurrence in each area.

used to describe the snowpack status classes are subjective and cannot be quantified precisely and uniformly over the content. Nevertheless, we provide a physical meaning of these terms, back upon recent detailed theoretical analyses from Colliander et al. (2022); Picard et al. (2022): "partial/full" distinguishes when less/more than 80 % of the pixel has meltwater, "surface"

indicates wet snow in the first 20 cm approximately and "depth" applies when the surface is dry and wet snow is below about 20 cm. The advantage of this qualitative classification is to gather the current knowledge of the sensitivity of each frequency beyond the separated single frequency binary indicators.

A critical issue is related to the validation. A strict validation of the classification is not possible because of the scarcity of in situ measurements in Antarctica, and also because measuring the surface and subsurface wetness using field techniques at

large scale is extremely difficult. The same limitation applies to the binary melt products widely used by the polar community. Following the strategy adopted by previous studies (e.g., Torinesi et al., 2003; Colliander et al., 2023), we addressed this issue by comparing our classification with surface air temperature from ERA5 and we showed that its variations support our physical interpretation (Sect. 4.1). This allows us to establish the temporal (Fig. 6) and spatial (Fig. 7) consistency of the classification to reproduce the most significant snowpack status variations.

The combination of observations acquired by different sensors and at different resolution is challenging (de Roda Husman et al., 2023) and introduces uncertainties into our classification. Combining observations at different frequencies inevitably results in mixing different spatial resolutions (the ground resolution is proportional to the wavelength for a given antenna size). Here, the most stringent difference is between AMSR2 (∼10 km at 37 GHz) and SMOS (∼40–70 km). The use of the enhanced-resolution SMOS brightness temperature product from Zeiger et al. (2024) provides an effective spatial resolution of ∼30 km



for SMOS, which helps reduce the sensor differences. Moreover, the definition of a quality flag allows us to mitigate this issue, by identifying non-physical signatures that may result from the different spatial representativeness of the observations. For instance, the signatures for which 19 and 37 GHz indicate full melt whereas 1.4 GHz indicates dry snow (signatures 60 and 62) are flagged as fair and may be related to this resolution issue.

     Some uncertainties are also related to the variability in penetration depth of each frequency. Defining the precise thickness
for detecting meltwater is challenging due to its dependency on several snow properties (snow temperature, density, and grain size) and on their vertical profile, which change over time. These properties indeed evolve throughout the melt season: the grain size increases due to wet metamorphism (Colbeck, 1982), and the density increases during the melt-refreeze cycles. As a consequence, the microwave penetration depth is likely to be greater at the beginning of the season than at the end. Despite this difficulty, rough estimates of this thickness were given for each frequency through theoretical analysis (e.g. Picard et al.,
2022) and empirical correlation (e.g. Colliander et al., 2022). Following them, we consider as an acceptable assumption to associate 37 GHz with the first centimetres of the snowpack (0-20 cm), 19 GHz with the first meters of the snowpack (1-2 m), and 1.4 GHz with depths exceeding 1 m and up to 10 m or more depending on the wet snow thickness (Picard et al., 2022; Leduc-Leballeur et al., 2020). However, more advanced modeling coupled with in situ measurements through the firn will be needed to refine the sensitivity knowledge of each frequency to depth-dependent liquid water.

Overall, despite these limitations mostly associated with the lack of in situ observations, the synthetic 10 snowpack classes are more user-friendly than the raw 64 dry-wet signatures derives from single-frequency binary indicators and permit further physical interpretation compared to existing binary melt products.

## 5    Conclusions

Dry-wet snow status in Antarctica has been explored over the period 2012-2023 using SMOS and AMRS2 observations. By
combining several frequencies (1.4, 19 and 37 GHz), day and night observations, and a binary indicator on melt coverage of each pixel, we are able to deliver qualitative information on the melt processes and liquid water distribution in the snowpack. Despite some subjectivity in this process and the lack of in situ measurements for validation, the ordering of the classes from "dry" (class 0) to "full melting" (class 9) is in agreement with ERA5 skin temperature variations at large scale and over multiple years. The resulting dataset shows the expected physical behaviours of the melt evolution throughout a season, with brief events
at the start and end of the season, full melting in December-January coinciding with the peak of melt extent, and persistent liquid water at depth without surface melting during the decreasing period of melt occurrences. In addition, the interannual variability in the Antarctic Peninsula indicates that years with more intense melting at the peak of the melt season exhibit persistent water at depth at the end. This new dataset provides, in a concise manner, the benefit of the current multi-frequency knowledge and opens perspectives to further explore the climate of Antarctic coastal areas.

In the future, we suggest further work to improve the robustness of the detection at the highest frequency used here (37 GHz) and potentially try to exploit higher and other intermediate frequencies commonly available (e.g., 89, 6 and 10 GHz) to refine the retrieved information on the vertical profile of the liquid water. Extending the 11-year timeseries with the upcoming JAXA



AMSR3 and the future ESA Copernicus Imaging Microwave Radiometer (CIMR) will offer a continuous and multi-decade climate perspective.

*Data availability.* The single frequency dry-wet snow indicators and the classification will be available on easydata.earth (as soon as validated).

*Author contributions.* MLL, GP: Conceptualization, Methodology. All authors: Formal analysis, Writing.

*Competing interests.* The authors declare that they have no conflict of interest.

*Acknowledgements.* This study benefited from European Space Agency support through the 4DAntarctica project (ESA/AO/1-9570/18/I-
DT).

**Appendix A: Dry-wet signature descriptions**





| N. | Dry-wet signature | | | | | | Description | QF | Class |
|---|---|---|---|---|---|---|---|---|---|
| | 80% wet | 19 ASC | 19 DSC | 37 ASC | 37 DSC | 1.4 DAY | | | |
| 0 | 0 | 0 | 0 | 0 | 0 | 0 | Dry | good | 0 |
| 1 | 0 | 0 | 0 | 0 | 0 | 1 | Deep liquid water | good | 1 |
| 2 | 0 | 0 | 0 | 0 | 1 | 0 | Weak night surface melting | fair | 0 |
| 3 | 0 | 0 | 0 | 0 | 1 | 1 | Weak night surface melting with deep liquid water | fair | 1 |
| 4 | 0 | 0 | 0 | 1 | 0 | 0 | Weak surface melting with night surface refreezing | fair | 0 |
| 5 | 0 | 0 | 0 | 1 | 0 | 1 | Weak surface melting with night surface refreezing and deep liquid water | fair | 1 |
| 6 | 0 | 0 | 0 | 1 | 1 | 0 | Weak surface melting | fair | 0 |
| 7 | 0 | 0 | 0 | 1 | 1 | 1 | Weak surface melting with deep liquid water | fair | 1 |
| 8 | 0 | 0 | 1 | 0 | 0 | 0 | Nighttime melting | good | 6 |
| 9 | 0 | 0 | 1 | 0 | 0 | 1 | Nighttime melting with deep liquid water | good | 6 |
| 10 | 0 | 0 | 1 | 0 | 1 | 0 | Nighttime melting and night surface melting | good | 6 |
| 11 | 0 | 0 | 1 | 0 | 1 | 1 | Nighttime melting and night surface melting with deep liquid water | good | 6 |
| 12 | 0 | 0 | 1 | 1 | 0 | 0 | Nighttime melting and daytime weak surface melting | good | 6 |
| 13 | 0 | 0 | 1 | 1 | 0 | 1 | Nighttime melting and daytime weak surface melting with deep liquid water | good | 6 |
| 14 | 0 | 0 | 1 | 1 | 1 | 0 | Nighttime melting and surface melting | good | 6 |
| 15 | 0 | 0 | 1 | 1 | 1 | 1 | Nighttime melting and surface melting with deep liquid water | good | 6 |
| 16 | 0 | 1 | 0 | 0 | 0 | 0 | Daytime melting with night refreezing | good | 2 |
| 17 | 0 | 1 | 0 | 0 | 0 | 1 | Daytime melting with night refreezing | good | 2 |
| 18 | 0 | 1 | 0 | 0 | 1 | 0 | Daytime melting with night refreezing but night surface melting | fair | 2 |
| 19 | 0 | 1 | 0 | 0 | 1 | 1 | Daytime melting with night refreezing but night surface melting | fair | 2 |
| 20 | 0 | 1 | 0 | 1 | 0 | 0 | Daytime melting with night refreezing | good | 2 |
| 21 | 0 | 1 | 0 | 1 | 0 | 1 | Daytime melting with night refreezing | good | 2 |
| 22 | 0 | 1 | 0 | 1 | 1 | 0 | Daytime melting with night refreezing but night surface melting | good | 2 |
| 23 | 0 | 1 | 0 | 1 | 1 | 1 | Daytime melting with night refreezing but night surface melting | good | 2 |
| 24 | 0 | 1 | 1 | 0 | 0 | 0 | Presence of liquid water | good | 4 |
| 25 | 0 | 1 | 1 | 0 | 0 | 1 | Presence of liquid water | good | 4 |
| 26 | 0 | 1 | 1 | 0 | 1 | 0 | Presence of liquid water and night surface melting | good | 4 |
| 27 | 0 | 1 | 1 | 0 | 1 | 1 | Presence of liquid water and night surface melting | good | 4 |
| 28 | 0 | 1 | 1 | 1 | 0 | 0 | Presence of liquid water with night surface refreezing | good | 3 |
| 29 | 0 | 1 | 1 | 1 | 0 | 1 | Presence of liquid water with night surface refreezing | good | 3 |
| 30 | 0 | 1 | 1 | 1 | 1 | 0 | Melting | good | 5 |
| 31 | 0 | 1 | 1 | 1 | 1 | 1 | Melting | good | 5 |
| 32 | 1 | 0 | 0 | 0 | 0 | 0 | Dry with high TB19 | poor | -1 |
| 33 | 1 | 0 | 0 | 0 | 0 | 1 | Dry with high TB19 | poor | -1 |
| 34 | 1 | 0 | 0 | 0 | 1 | 0 | Dry with high TB19 | poor | -1 |
| 35 | 1 | 0 | 0 | 0 | 1 | 1 | Dry with high TB19 | poor | -1 |
| 36 | 1 | 0 | 0 | 1 | 0 | 0 | Dry with high TB19 | poor | -1 |
| 37 | 1 | 0 | 0 | 1 | 0 | 1 | Dry with high TB19 | poor | -1 |
| 38 | 1 | 0 | 0 | 1 | 1 | 0 | Melting | fair | 5 |
| 39 | 1 | 0 | 0 | 1 | 1 | 1 | Melting | fair | 5 |
| 40 | 1 | 0 | 1 | 0 | 0 | 0 | Presence of liquid water | fair | 4 |
| 41 | 1 | 0 | 1 | 0 | 0 | 1 | Presence of liquid water | fair | 4 |
| 42 | 1 | 0 | 1 | 0 | 1 | 0 | Presence of liquid water | fair | 4 |
| 43 | 1 | 0 | 1 | 0 | 1 | 1 | Presence of liquid water | fair | 4 |
| 44 | 1 | 0 | 1 | 1 | 0 | 0 | Melting with surface night refreezing | fair | 3 |
| 45 | 1 | 0 | 1 | 1 | 0 | 1 | Melting with surface night refreezing | fair | 3 |
| 46 | 1 | 0 | 1 | 1 | 1 | 0 | Melting | fair | 5 |
| 47 | 1 | 0 | 1 | 1 | 1 | 1 | Melting | fair | 5 |
| 48 | 1 | 1 | 0 | 0 | 0 | 0 | Daytime full melting with night refreezing | good | 7 |
| 49 | 1 | 1 | 0 | 0 | 0 | 1 | Daytime full melting with night refreezing | good | 7 |
| 50 | 1 | 1 | 0 | 0 | 1 | 0 | Daytime full melting with night refreezing but night surface melting | poor | 7 |
| 51 | 1 | 1 | 0 | 0 | 1 | 1 | Daytime full melting with night refreezing but night surface melting | poor | 7 |
| 52 | 1 | 1 | 0 | 1 | 0 | 0 | Daytime full melting with night refreezing | good | 7 |
| 53 | 1 | 1 | 0 | 1 | 0 | 1 | Daytime full melting with night refreezing | good | 7 |
| 54 | 1 | 1 | 0 | 1 | 1 | 0 | Daytime full melting with night refreezing but night surface melting | poor | 7 |
| 55 | 1 | 1 | 0 | 1 | 1 | 1 | Daytime full melting with night refreezing but night surface melting | poor | 7 |
| 56 | 1 | 1 | 1 | 0 | 0 | 0 | Presence of liquid water but with TB1.4=0 | fair | 4 |
| 57 | 1 | 1 | 1 | 0 | 0 | 1 | Presence of liquid water | good | 4 |
| 58 | 1 | 1 | 1 | 0 | 1 | 0 | Full melting but daytime surface refreezing and TB1.4=0 | poor | 4 |
| 59 | 1 | 1 | 1 | 0 | 1 | 1 | Full melting but daytime surface refreezing | good | 4 |
| 60 | 1 | 1 | 1 | 1 | 0 | 0 | Full melting with night surface refreezing with TB1.4=0 | fair | 8 |
| 61 | 1 | 1 | 1 | 1 | 0 | 1 | Full melting with night surface refreezing | good | 8 |
| 62 | 1 | 1 | 1 | 1 | 1 | 0 | Full melting with TB1.4=0 | fair | 9 |
| 63 | 1 | 1 | 1 | 1 | 1 | 1 | Full melting | good | 9 |

**Figure A1.** The 64 dry-wet signatures with their physical meaning, their associated quality flag (QF) and assigned class.



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
