# Peer review of "Empirical classification of dry-wet snow status in Antarctica using multi-frequency passive microwave observations"

_EGUsphere, 2025_

## Author Comment (AC1)

**Empirical classification of dry-wet snow status in Antarctica using multi-frequency passive microwave observations**

Marion Leduc-Leballeur1, Ghislain Picard2, Pierre Zeiger2, and Giovanni Macelloni1

1Institute of Applied Physics "Nello Carrara", National Research Council, 50019 Sesto Fiorentino, Italy

**Correspondence:** Marion Leduc-Leballeur (m.leduc@ifac.cnr.it)

**Reviewer comment 1**

https://doi.org/10.5194/egusphere-2025-732-RC1

The manuscript presents a multi-depth snowpack status classification scheme for Antarctica, utilizing multi-frequency spaceborne microwave radiometry. The paper is clearly written, and the subject matter aligns well with the scope and interests of the journal.

The use of the full spectrum of passive microwave radiometry for ice sheet melt detection is a relatively understudied area, which has fortunately begun to receive more attention in recent years. This study contributes meaningfully to that growing body of work.

While the approach may be seen as a preliminary or "low-hanging fruit" analysis, the authors' qualitative investigation of microwave signal behavior in relation to melt evolution represents an essential foundational step. This work is critical for advancing future research aimed at extracting more detailed and quantitative insights from remote sensing data.

I consider this paper a valuable contribution to the field and recommend it for publication following minor revisions:

We would like to thank the reviewer for the time devoted to this review. Please, find our answers in blue to the comments hereinafter.

Line 43: Please change "among" to "amount."

Done.

Line 44: Add citations to relevant recent work, such as: Naderpour et al. (2020), Mousavi et al. (2022), Hossan et al. (2024), Moon et al. (2024)

Thank you for the suggestion of these recent works. We will add as follows:

"(...) The algorithms previously developed for higher frequencies have proven effective at L-band as well (Leduc-Leballeur et al., 2020; Mousavi et al., 2022). (...) This unique characteristic has recently been exploited in Greenland to detect perennial firn aquifers with SMAP (Miller et al., 2020) and to estimate the total liquid water amount (Naderpour et al., 2020; Houtz et al., 2021; Hossan et al., 2025; Moon et al., 2025). This new perspective highlights the different information retrieved depending on the frequency."

<sup>2UGA, CNRS, Institut des Géosciences de l'Environnement (IGE), UMR 5001, 38041 Grenoble, France

Line 125: The choice to define the new melt season starting in mid-autumn is not intuitive. As Figure 5 suggests, late-season melt events may be incorrectly attributed to the following melt season. Please clarify the rationale or consider adjusting the definition.

The period April to April is only used by the detection algorithm to compute the winter statistics (mean and standard deviation). The algorithm is designed to remove all melt events over this period, whenever they occur, without considering a beginning or an end of a season. Any period of 12 months could be used, but it is preferred to have a single winter in this period.

On the other hand, for the presentation and the discussion of the results, we computed all the seasonal statistics and discussed the interannual variability throughout the article using the period from September N to June N+1, as illustrated in Figure 5 of the article.

For clarification, we propose reformulating the sentence in Section 3.1, L125:

"The algorithm determines an optimal brightness temperature threshold in every grid cell and time period from 1 April year N to 31 March year N+1. It considers that any acquisition of brightness temperature higher than this threshold indicates wet snow."

40 Lines 139–140: Please cite additional supporting literature, such as: Macelloni et al. (2011), Montomoli et al. (2022)

Thank you for your suggestion. We noted that Macelloni et al. (2011); Montomoli et al. (2022) refer to the L-band frequency while this paragraph address the 19 GHz frequency. Thus, it seems to us that the already cited Picard et al., 2022 and Colliander et al., 2022 support well the use of vertical polarization to detect liquid water with 19 GHz.

Lines 179–180: This statement may oversimplify the L-band response. The response depends on the amount of active melting and liquid water accumulation. Please clarify this dependency here.

We agree that the response depends on the amount of liquid water. We propose reformulating the sentence as follows:

"Note that when liquid water is present in a sufficient amount to be detected at a given level (e.g. at the surface when active melting is occurring), radiation emanating from below is blocked, and no information on the dry-wet status can be detected under the highest level. This blocking effect depends on the amount of liquid water, the thickness of the wet layer, and of the frequency. However, in practice, if liquid water is detected at a given high frequency, it is unreliable to exploit lower frequencies to determine the status below the upper wet level (Picard et al., 2022)."

Line 197: Consider replacing "coherent" with "consistent" for improved clarity.

Done.

Line 235: Correct the grammar: "this possibilities" should be "these possibilities."

55 Done.

35

Lines 297–298: Consider emphasizing the adequate accumulation of liquid water, rather than just the depth, as the key factor influencing the observed response.

We will reformulate in: "No wet status is detected at 1.4 GHz, suggesting that the amount of liquid water is low and does not affect the snowpack at depth."

**60 References**

65

70

80

85

- Hossan, A., Colliander, A., Vandecrux, B., Schlegel, N.-J., Harper, J., Marshall, S., and Miller, J. Z.: Retrieval and validation of total seasonal liquid water amounts in the percolation zone of the Greenland ice sheet using L-band radiometry, The Cryosphere, 19, 4237–4258, https://doi.org/10.5194/tc-19-4237-2025, 2025.
- Houtz, D., Mätzler, C., Naderpour, R., Schwank, M., and Steffen, K.: Quantifying Surface Melt and Liquid Water on the Greenland Ice Sheet using L-band Radiometry, Remote Sensing of Environment, 256, 112 341, https://doi.org/10.1016/j.rse.2021.112341, 2021.
- Leduc-Leballeur, M., Picard, G., Macelloni, G., Mialon, A., and Kerr, Y. H.: Melt in Antarctica derived from Soil Moisture and Ocean Salinity (SMOS) observations at L band, The Cryosphere, 14, 539–548, https://doi.org/10.5194/tc-14-539-2020, 2020.
- Macelloni, G., Brogioni, M., Pettinato, S., Crepaz, A., and Zasso, R.: Technical Support for the Deployment of an L-band Radiometer at Concordia Station During DOMEX-2 and Data Analysis. Final Report. Version 2.0., Tech. rep., European Space Agency Study Contract Reports, https://earth.esa.int/eogateway/documents/20142/37627/DOMEX-2-Final-Report.pdf, 2011.
- Miller, J. Z., Long, D. G., Jezek, K. C., Johnson, J. T., Brodzik, M. J., Shuman, C. A., Koenig, L. S., and Scambos, T. A.: Brief communication: Mapping Greenland's perennial firn aquifers using enhanced-resolution L-band brightness temperature image time series, The Cryosphere, 14, 2809–2817, https://doi.org/10.5194/tc-14-2809-2020, 2020.
- Montomoli, F., Brogioni, M., Macelloni, G., Leduc-Leballeur, M., Baldi, M., Martin-Neira, M., and Casal, T. G. D.: Long Term L-Band
  Brightness Temperature of the DOMEX-3 Experiment: Improvement of Absolute Calibration and Data Analysis, in: IGARSS 2022 2022
  IEEE International Geoscience and Remote Sensing Symposium, pp. 7367–7370, https://doi.org/10.1109/igarss46834.2022.9883561, 2022.
  - Moon, T., Harper, J., Colliander, A., Hossan, A., and Humphrey, N.: L-Band Radiometric Measurement of Liquid Water in Greenland's Firn: Comparative Analysis with In Situ Measurements and Modeling, Annals of Glaciology, pp. 1–21, https://doi.org/10.1017/aog.2025.10012, 2025.
  - Mousavi, M., Colliander, A., Miller, J., and Kimball, J. S.: A Novel Approach to Map the Intensity of Surface Melting on the Antarctica Ice Sheet Using SMAP L-Band Microwave Radiometry, IEEE Journal of Selected Topics in Applied Earth Observations and Remote Sensing, 15, 1724–1743, https://doi.org/10.1109/jstars.2022.3147430, 2022.
  - Naderpour, R., Houtz, D., and Schwank, M.: Snow wetness retrieved from close-range L-band radiometry in the western Greenland ablation zone, Journal of Glaciology, 67, 27–38, https://doi.org/10.1017/jog.2020.79, 2020.
  - Picard, G., Leduc-Leballeur, M., Banwell, A. F., Brucker, L., and Macelloni, G.: The sensitivity of satellite microwave observations to liquid water in the Antarctic snowpack, The Cryosphere, 16, 5061–5083, https://doi.org/10.5194/tc-16-5061-2022, 2022.

---

## Author Comment (AC2)

**Empirical classification of dry-wet snow status in Antarctica using multi-frequency passive microwave observations**

Marion Leduc-Leballeur1, Ghislain Picard2, Pierre Zeiger2, and Giovanni Macelloni1

Correspondence: Marion Leduc-Leballeur (m.leduc@ifac.cnr.it)

**Reviewer comment 2**

https://doi.org/10.5194/egusphere-2025-732-RC2

This work proposes a novel synthesis of binary melt determinations from several different passive microwave sources, as well as accounting for varying diurnal observation timings, in order to assert features about the melt state of Antarctic firn. The authors propose a classification system to relate observations from 3 different satellites, and 6 different melt detection algorithms, into a set of categories for the spatial and diurnal variation of liquid water in a firn column. These classes go beyond what has previously been derived from passive microwave observation synthesis, and this work provides a valuable advance in passive microwave observation analysis of melt. This work provides a comparison with ERA5 reanalysis skin temperatures to give confidence that the durnality and spatial distribution of surface melt in this work at least generally reflects real world patterns, which is a reasonable approach for passive microwave melt analysis. Aside from several minor questions below, I believe that this work is suitable for publication in this journal.

We would like to thank the reviewer for the time devoted to this review. Please, find our answers in blue to the comments hereinafter.

- 15 My two biggest questions have to do with the new melt detection techniques presented in this work.
  - 1) A new method for melt detection at 37 GHz is proposed here, and given its novelty I would like to see a little additional discussion about this method. In particular, I would like to see an explanation for the rationale of using a running mean of recent dry brightness temperature values and a 1-sigma increase to detect melt. Additionally, I am curious to see a short discussion of how well the method performs, or at least how well it agrees with other melt detection results.
- Thank you for this comment. We propose to better describe the new melt detection method used for 37 GHz and split Section 3.1 into subsections dedicated to each indicator to improve clarity: 3.1.1 19 GHz-based dry-wet snow indicator, 3.1.2 1.4 GHz-based dry-wet snow indicator, 3.1.3 37 GHz-based dry-wet snow indicator, 3.1.4 Full-partial melting pixel indicator. We propose adding the following explanation in L157:

"Nonetheless, the Torinesi et al. (2003) method previously used for melt detection at 19 GHz and 1.4 GHz is inadequate.

5 Firstly, 37 GHz is strongly affected by snow metamorphism in the first centimeters and some large and rapid brightness temper-

<sup>1Institute of Applied Physics "Nello Carrara", National Research Council, 50019 Sesto Fiorentino, Italy

<sup>2UGA, CNRS, Institut des Géosciences de l'Environnement (IGE), UMR 5001, 38041 Grenoble, France

ature variations may be related to change in snow grain size and density rather than melt (Brucker et al., 2011; Champollion et al., 2019). Even at Dome C where no melt occurs some rapid variations of 5-10 K can be observed during winter at 37 GHz (Brucker et al., 2011). Moreover, the large brightness temperature seasonal cycle at 37 GHz makes it difficult to use the Torinesi et al. (2003) method, which is based on the hypothesis of moderate variations of dry brightness temperature. Secondly, the seasonal melt, refreezing cycles and precipitations change the ice properties at the surface and can generate strong variations in brightness temperature at 37 GHz. A constant threshold over the April year N to March year N+1 period, as used at lower frequencies (Torinesi et al., 2003), is unadapted.

As we need to distinguish the brightness temperature variations related to rapid changes in the ice surface properties, such as grain size or density, from those related to the liquid water presence, we propose to use a running mean instead of a fixed annual threshold as in Torinesi et al. (2003). A new threshold definition was adopted:  $T_{37} = M_{37} + \sigma_{37}$  where  $M_{37}$  is the 5-day moving mean timeseries of the brightness temperatures when the 19 GHz indicator is dry and  $\sigma_{37}$  is its standard deviation between 1 April year N to 31 March year N+1. The  $M_{37}$  timeseries is then linearly interpolated to fill the gaps when the 19 GHz indicator is wet.  $T_{37}$  is computed from the brightness temperature in vertical polarisation acquired at ascending passes and subsequently applied to both ascending and descending passes.

40 Figure 1 shows an example of the new threshold definition for one grid cell from September 2015 to June 2016, with which 30 wet snow days were identified. For comparison, the Torinesi et al. (2003) threshold applied to 37 GHz timeseries detects 20 melt days, and the main differences are observed from mid-January to deb-February. During this period, 37 GHz brightness temperature has strong variations of more than 40 K in one day, which could be attributed to liquid water. Moreover, the ERA daily maximum temperature around 270 K also suggest the possibility of wet snow. In general, over the 2012-2023 period, the 37 GHz dry-wet indicator computed from Torinesi et al. (2003) and this study are in agreement in 99.1% of the cases when at least one wet snow day is detected by one of the two methods. The Torinesi et al. (2003) indicator detected wet (dry) snow whereas the indicator of this study detected dry (wet) in 0.5% (0.4%) of cases.

Finally, note that this new threshold still exhibits a strong sensitivity to brightness temperature variations, leading to occasional unexpected melt detection during winter (e.g. 296 pixels in July-August on average, i.e. 0.13 % of the total wet days detected with 37 GHz). These false alarms underscore that melt detection at this frequency remains difficult and subject to uncertainties."

**Figure 1.** Brightness temperature at 37 GHz (purple) at 67.15°S, 84.13°E on West ice shelf in 2015/16 and the thresholds from Torinesi et al. (2003) (dashed) and from this study (dotted). ERA5 daily maximum skin temperature (grey).

2) A method for identifying if 80% of a 19 GHz pixel is melting is introduced. This method uses the difference between dry and wet snow brightness temperatures to produce a threshold value. I am curious, why did you use a single value for 19 GHz Tdry for all surfaces? I would expect the dry snow 19 GHz value to vary spatially as it is a function of temperature profile, grain size, and ice lenses in a snowpack.

55

60

We agree that the dry snow brightness temperature at 19 GHz depends on snow grain size, ice lenses and this generates spatial variations. Our hypothesis was that when snow becomes wet, brightness temperature tends to be independent of the grain size because the liquid water presence masked out the sensitivity to the other parameters and to the layers under the wet horizon.

However, we noted that using a  $T_{\rm dry}$  spatially and periodically varying, computed as in Torinesi et al. (2003), provides  $T_{80\%}$  ranges between 248 K and 270 K with a median of 261 K. Figure 2 shows the difference of the percentage of occurrences for each signature. In total, about 4% of the wet occurrences are affected by a signature change. The most impacted signature is the signature 32 which is composed by the  $T_{80\%}$  at one and all the others indicator at zero. This signature is defined as invalid and excluded from the classification because the  $T_{80\%}$  indicator and the 19 GHz in ascending pass indicator are in opposition. Using an adaptive  $T_{80\%}$  strongly reduces the occurrence of this absurd signature (from 1.9% to 0.3%), enabling 1.6% of occurrences that would be excluded by a constant  $T_{80\%}$  to be included in the classification.

We propose to use an adaptive  $T_{80\%}$  to keep the consistency with the other indicators, which all use adaptive thresholds. Note that, no sharp change has been observed in the analysis performed in the article by using an adaptive  $T_{80\%}$ . We will add the following description in the text and update figures and numbers in the article: " $M_{dry}$  is the mean brightness temperature of the dry snow computed between 1 April year N to 31 March year N+1, as described in Section 3.1.1 from Torinesi et al. (2003)."

Figure 2. Difference of the percentage of occurrences for each signature.

A few specific comments by line:

Lines 141-142: You apply a bound of 20 to 35 K for the min and maximum threshold when applying the Torinesi method on 19V GHz data. How often are these bounds used? If they are required often, then how sensitive is the melt detection to those bounds?

For the 19 GHz indicator, the upper (and lower) bounds are used for 0.3% (and 78%) of pixels that experienced wet snow for at least one day during the 12-year period. The upper bound was used very rarely, only in some marginal ice shelf areas. To assess the effect of the lower bound, we computed statistics using pixels where wet snow is detected at least 1 day over the 2012-2023 period without applied a lower bound. Figure 3 presents the annual mean percentage of pixels for which the lower bound has been activated and the associated cumulative melting surface (CMS). CMS is defined as the number of wet pixels for an entire period over which the threshold is computed (i.e. April year N to March year N+1) and an entire region multiplied by the surface of a pixel (12.5 x 12.5 km). Using a lower bound of 15 K reduces by 23 % the use of this bound, and induces a increase in CMS of only 6%, which is relatively small. We also observed that, using a 20 K lower bound reduces by 6% the wet snow detection during winter (July-August-September period) and of 18% the wet snow detection at surface elevation higher than 1700 m.

Note also that theoretical analysis over the Antarctic Plateau suggested that variations of brightness temperature in vertical polarisation at 19 GHz lower than 20 K are probably related to snow surface metamorphism, given the 19 GHz sensitivity to variations in grain size (Brucker et al., 2011).

All of this supports the use of a lower bound of 20 K. This choice results in a conservative dry-wet snow detection that tends to reduce false alarm relative to undetected events.

We propose adding in the section: "The upper (and lower) bounds are used for 0.3 % (and 78 %) of pixels that experienced wet snow for at least one day during the 2012-2023 period. The upper bound was used very rarely, only in some marginal ice shelf areas. Sensitivity analysis showed that using a lower bound of 20 K instead of 15 K reduces the detection of wet snow by

**Figure 3.** Annual mean of (left) the percentage of pixels for which the lower bound has been activated and (right) the associate cumulative melting surface.

6 % in winter (the July-September period) and by 18 % at surface elevation higher than 1700 m. This choice of a lower bound of 20 K results in a conservative dry-wet snow detection that tends to reduce false alarm relative to undetected events."

**Line 150: as with my previous comment, how often are these bounds used?**

For the 1.4 GHz indicator, the upper (and lower) bounds are used for 1.1% (and 96.5%) of pixels that experienced wet snow for at least one day during the 12-year period. The upper bound was used very rarely. To assess the effect of the lower bound, we computed statistics using pixels where wet snow is detected at least 1 day over the 2012-2023 period without applied a lower bound. Figure 4 presents the annual mean percentage of pixels for which the lower bound has been activated and the associated cumulative melting surface (CMS). CMS is defined as the number of wet pixels for an entire period over which the threshold is computed (i.e. April year N to March year N+1) and an entire region multiplied by the surface of a pixel (12.5 x 12.5 km). Using a lower bound of 5 K reduces by 58 % the use of this bound, and induces a increase in CMS of 45 %. We observed that this large variation is related to the fact that, using a 10 K lower bound reduces by 85 % the wet snow detection during winter(July-August-September period) and by 89 % the wet snow detection at surface elevation higher than 1700 m.

Note also that theoretical analysis over the Antarctic Plateau suggested that variations of brightness temperature in horizontal polarisation at 1.4 GHz lower than 5 K are probably related to snow surface metamorphism (Brucker et al., 2014; Leduc-Leballeur et al., 2017).

All of this supports the use of a lower bound of 10 K. This choice results in a conservative dry-wet snow detection that tends to reduce false alarm relative to undetected events.

We propose adding in the section: "The upper (and lower) bounds are used for 1.1 % (and 96.5 %) of pixels that experienced wet snow for at least one day during the 2012-2023 period. The upper bound was used very rarely. Sensitivity analysis showed that using a lower bound of 10 K instead of 5 K reduces the detection of wet snow by 85 % in winter (the July-September

**Figure 4.** Annual mean of (left) the percentage of pixels for which the lower bound has been activated and (right) the associate cumulative melting surface.

period) and by 89 % at surface elevation higher than 1700 m. The 10 K lower bound enables a significant reduction in false alarms."

Lines 150-153: Just to be clear, does "filtered out" mean that these pixels are given no result for the year, or listed as no melt all year.

We will clarify the text as: "Moreover, pixels with a standard deviation of brightness temperature in vertical polarisation lower than 2.8 K from 1 April year N to 31 March year N+1 are marked as dry for this period (Leduc-Leballeur et al., 2020)."

Lines 158-159: Is the M37 value calculated from the 5 most recent days that the snow was dry at 19 GHz? The text was unclear what the algorithm does if there are more than 5 days of 19 GHz melting.

This information is indeed missing in the method description. We propose to add:

"The  $M_{37}$  timeseries is then linearly interpolated to fill the gaps when the 19 GHz indicator is wet."

Table 1: I am interested to see the occurrence rate of each of these classes or signatures listed somewhere, which could go in this table or Figure 1A in the appendix. Alternatively, in Figure 6 a second row could be added plotting the relative prevalence of each melt class by day of year.

For the 64 dry-wet signature, the relative occurrence rates are presented in Figure 2. For the snowpack classes, we agree to follow your suggestion, and we propose adding an inset plot in the Figure 6 of the article as in Figure 5.

**Figure 5.** Daily extent of each snowpack status over Antarctica from September to June on average over 2012-2023. The percentage of occurrences of each class over this period is shown in the upper left inset plot.

**130 References**

135

Brucker, L., Picard, G., Arnaud, L., Barnola, J.-M., Schneebeli, M., Brunjail, H., Lefebvre, E., and Fily, M.: Modeling time series of microwave brightness temperature at Dome C, Antarctica, using vertically resolved snow temperature and microstructure measurements, Journal of Glaciology, 57, 171–182, https://doi.org/10.3189/002214311795306736, 2011.

Brucker, L., Dinnat, E. P., Picard, G., and Champollion, N.: Effect of snow surface metamorphism on Aquarius L-band radiometer observations at Dome C, Antarctica, IEEE Trans. Geosci. Remote Sens., 52, 7408–7417, https://doi.org/10.1109/TGRS.2014.2312102, 2014.

Champollion, N., Picard, G., Arnaud, L., Lefebvre, E., Macelloni, G., Rémy, F., and Fily, M.: Marked decrease in the near-surface snow density retrieved by AMSR-E satellite at Dome C, Antarctica, between 2002 and 2011, The Cryosphere, 13, 1215–1232, https://doi.org/10.5194/tc-13-1215-2019, 2019.

Leduc-Leballeur, M., Picard, G., Macelloni, G., Arnaud, L., Brogioni, M., Mialon, A., and Kerr, Y.: Influence of snow sur-140 face properties on L-band brightness temperature at Dome C, Antarctica, Remote Sensing of Environment, 199, 427–436, https://doi.org/10.1016/j.rse.2017.07.035, 2017.

Leduc-Leballeur, M., Picard, G., Macelloni, G., Mialon, A., and Kerr, Y. H.: Melt in Antarctica derived from Soil Moisture and Ocean Salinity (SMOS) observations at L band, The Cryosphere, 14, 539–548, https://doi.org/10.5194/tc-14-539-2020, 2020.

Torinesi, O., Fily, M., and Genthon, C.: Variability and trends of the summer melt period of Antarctic ice margins since 1980 from microwave sensors, Journal of Climate, 16, 1047–1060, 2003.